# Untangling Coastal Diversity: How Habitat Complexity Shapes Demersal and Benthopelagic Assemblages in NW Iberia

**Marisa A. Gomes** [1,2,*] , **Catarina M. Alves** [1,2], **Fábio Faria** [1], **Jesus S. Troncoso** [2] **and Pedro T. Gomes** [1]

1 Centre of Molecular and Environmental Biology (CBMA), Department of Biology, University of Minho, Campus de Gualtar, 4710-057 Braga, Portugal; catarinamacedoalves@gmail.com (C.M.A.); fabio_faria15@hotmail.com (F.F.); pagomes@bio.uminho.pt (P.T.G.)

2 Department of Ecology and Animal Biology, ECOCOST Lab, Marine Research Centre (CIM-UVIGO), University of Vigo, Campus Lagoas-Marcosende, 36310 Vigo, Spain; troncoso@uvigo.es

* Correspondence: marisantunesgomes@gmail.com

**Abstract:** Understanding species–habitat relationships is essential for ecosystem-based conservation. This study explores the significance of habitat characteristics and complexity for demersal and benthopelagic communities within a patchwork of coastal habitats, including rocky seabed, macroalgae formations, sandy bottoms, and a combination of rock and sand areas. Species and habitats were surveyed along the north-west (NW) Iberian continental shelf area of Viana do Castelo using baited remote underwater video stations (BRUVS). We found significant differences ($p < 0.05$) in species assemblages across habitats, with rocky substrates showing the highest diversity and abundance. Sand habitats showed the lowest species richness and abundance, underscoring the importance of habitat complexity to support marine life. Our study also emphasises the role of specific species in shaping the communities, identifying key species such as *Trisopterus luscus*, *Diplodus vulgaris*, and *Ctenolabrus rupestris* as the three most abundant in the region and significant contributors to the observed dissimilarities between habitats. By elucidating the impact of habitat complexity on marine life, our results offer essential baseline data, which serve as a kick-start point to inform sustainable management and conservation strategies for the long-term health and productivity of these vital ecological systems in the North-East Atlantic.

**Keywords:** habitat complexity; coastal conservation; marine assemblages; North-East Atlantic; baited remote underwater video system (BRUVS)

## 1. Introduction

Coastal habitats contribute significantly to the life cycles and population dynamics of different communities [1] and play a vital role in demersal and benthopelagic species, which serve as essential nurseries and feeding grounds [2–4]. Demersal and benthopelagic assemblages are crucial assets for coastal populations and depend, in some way, on the habitat structure for shelter, food, and reproduction [5]. The unsustainable exploitation of demersal and benthopelagic species, including declines in commercially important species [4,6], has been a growing concern. In the stock management of this fauna, the fishing effort control focuses mainly on the reproductive potential, closed seasons, size limits, catch limits, and gear restrictions [7,8]. Despite the comprehensive approach to the stock management of fauna, the existing measures predominantly overlook crucial factor–habitat characteristics—which is frequently underestimated. However, habitat integrity and structure are critical for many of these species, mainly for juvenile stages. The shallow coastal waters are essential for providing this integrity and structure, providing the conditions necessary for the development of the species [9]. Nevertheless, the areas needed by most of these species are under tremendous pressure induced by anthropogenic influences, and depending on the biological stage of the species, a specific habitat typology may be more importance than others [10]. The habitat mosaic heterogeneity across

coastal regions influences regional diversity by hosting diverse communities [11,12] and is a growing acknowledgement of the significance of this habitat's diversity for species populations [13–15]. Many species, for example, use multiple habitats during their life cycles and seasons [4,16], and these habitats are not isolated since they are interconnected through migration patterns, shared hydrological processes, sediment transportation, and nutrient flows [2].

Still, not only can the different combinations of habitats influence the species' presence, but the level of complexity of these habitats can also significantly influence the species themselves [17]. Habitats characterised by higher complexity can offer different microhabitats [18], presenting more shelter availability and food resources to different species with diverse life strategies. It is important to note that the extent of this habitat's complexity can also vary significantly within the same habitat category. Boulders of varying sizes, caves and crevices, and other features of the sea bottom play a crucial role in shaping coastal species assemblages, and the presence or absence of these features can influence marine communities in coastal areas. Many species prefer the occurrence of different crevice sizes or boulders because of their essential benefits, such as protection from predators or spaces for egg deposition, more than others [19–21]. At a regional scale in temperate regions, this complexity is one of the most influential physical factors (e.g., rocky reefs or coral reefs), increasing species richness, abundance, and biomass [22–24].

While an increasing body of research acknowledges the significance of habitat mosaic diversity, a deeper understanding of how various habitats and topographical complexity influence the assemblies of demersal and benthopelagic species along the north-west (NW) Iberian Coast is still required. In light of these aspects, developing foundational baselines that define and elucidate the influence of habitat and its complexity in species' distribution is crucial for an ecosystem-based approach to biodiversity conservation, regional fisheries management, and the spatial planning of coastal developments [25–27].

This study aims to fill this gap by gathering essential information on the NW Iberian coast in Viana do Castelo, Portugal. Baited remote underwater video stations (BRUVS) were utilised to survey species and their habitat associations in a multi-habitat approach. This method allows for visual and direct observation of species in their natural habitats. Although this coastal region features a variety of habitats, such as sandy bottoms, rocky reefs, and macroalgae formations, few studies have been conducted in the area. As a result, more data on the existing marine assemblages and their relationship with the habitat mosaic and topographic complexity must be collected. This creates an ideal opportunity to investigate these ecological dynamics.

The specific aims of this study were (i) to describe the habitat mosaic on the shallow coast (<30 m) of Viana do Castelo, (ii) to identify the demersal and benthopelagic assemblages in the area, (iii) to evaluate the influence of the bedrock nature in the support capacity of the habitat for demersal and benthopelagic organisms, and (v) to examine the role of the habitat mosaic and complexity in the distribution of the target species. Understanding these relationships is critical to establish marine conservation strategies, sustainably manage fisheries, and preserve the ecological integrity of coastal marine ecosystems.

## 2. Materials and Methods

### 2.1. Study Area

The study area is located on the NW Iberian continental shelf within the coastal region of Viana do Castelo, Portugal. This area extends to an approximate depth of 180 m and is influenced by two significant estuary systems: Rio Minho and Rio Lima [28]. Despite these estuary systems, the coastal region is characterised by a distinct lack of natural barriers, rendering it highly exposed to the direct impacts of coastal processes. Due to its vertical orientation (facing west), this coast is consistently exposed to different phenomena, such as high-wave energy regimes, predominantly facing energetic west and north-west, the southward surface "Portugal Current", the northward currents, upwelling, and intense wind patterns [29]. These factors significantly influence various sedimentary

processes, shaping the dynamics of the ecosystem. It is important to note that these coastal processes reduce underwater visibility and present challenging conditions for diving and nautical activities throughout the year [30,31]. Seasonal upwelling events are a prominent feature in the study area, with heightened consistency and intensity observed during the summer and early autumn [32], establishing it as one of the major upwelling systems [33]. These events are particularly prominent among the pivotal factors in enhancing primary productivity within the ecosystem [34]. The seabed in this region exhibits a diverse landscape, featuring rocky reefs resulting from plutonic and metamorphic outcrops [28,29], interspersed with sand banks, extensive rocky platforms, and kelps forest. These features create a diverse habitat mosaic with heterogeneous topography, which forms a complex coastal environment (Figure 1b). The characteristics of this area limit the methodologies and surveys that can be employed.

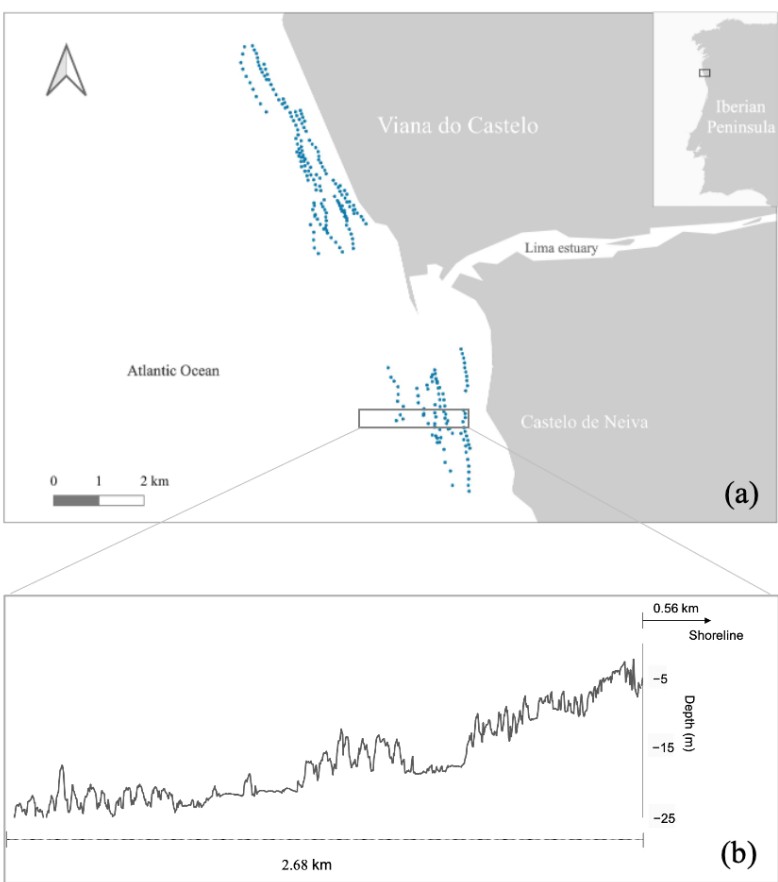

**Figure 1.** Map of the initial 246 BRUVs survey's location (blue dots) (**a**) and topographic profile of a section (**b**) of the Viana do Castelo (Portugal) coastal area. Topographic profile was created in QGIS based on COSMonline data.

### 2.2. BRUVS Apparatus and Sampling Protocol

The procedures in the field occurred between June and September of 2020 during the morning period (8:00–12:00 a.m.) work window time. Due to the intense North wind's characteristic in this area, which leads to challenging nautical conditions, sampling during the afternoon was not viable. Sampling locations (Figure 1a) were chosen based on the existing cartography of the bottom (e.g., COSMO, EMODnet, Navionics, and local nautical charts), complemented by additional scanning using multibeam sonar. Deployments were distributed across different habitat types to ensure the maximum coverage and balance, given the information available. The demersal and benthopelagic species were sampled using the 360° BRUVS designed by our team, specially developed to survey these communities in highly complex coastal environments. This design consists of a 360°

underwater camera (Insta 360 One X, Arashi Vision Inc., Irvine, CA, USA) screwed into a diving weight, forming the camera platform. The platform stays connected to a surface buoy and the bait bag is attached 50 cm from the camera (Figure 2a). BRUVS were deployed in 246 points along the study area and distributed between depths of around 4 and 25 metres (Figure 1a). Between the BRUVS deployments in the same round, a minimum distance of 150 m was adopted, and crushed sardines were used as bait. During the sampling months, the visibility and temperature of the water varied between 0.5 and 15 m and between 12° and 18 °C. Each deployment lasted approximately 60 min, including deployment, video recording, and BRUV recovery.

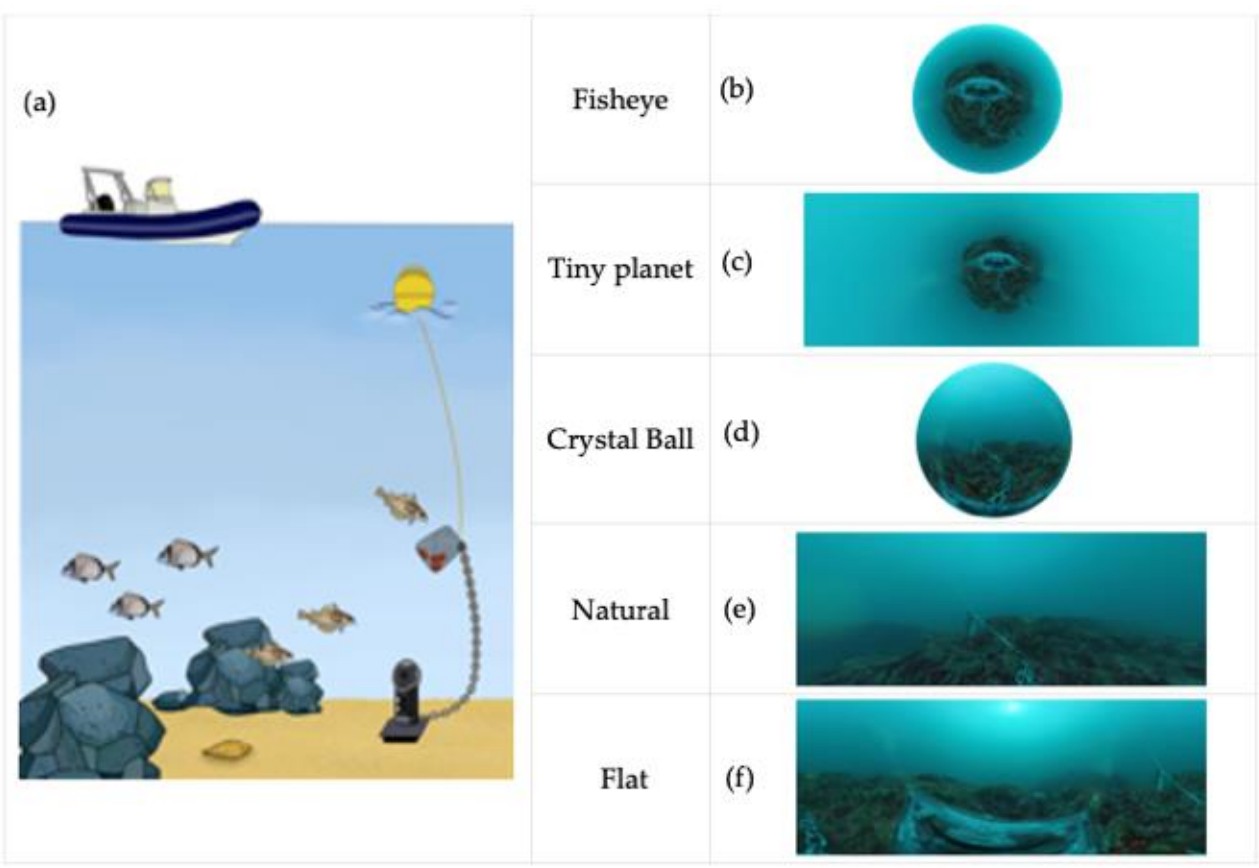

**Figure 2.** 360° BRUV scheme deployment (**a**) and multiview approaches. Fisheye view (**b**), tiny planet view (**c**), crystal ball (**d**), natural view (**e**), and flat view (**f**).

### 2.3. Video Analysis

The same observer performed the video analysis using Insta360 studio 2021 desktop editing software (version: 3.5.8) and the multiview options available in the software (Figure 2b–f). Videos were shot at 5760 × 2880 (5.7 k) at 30 fps. However, it is important to note that the spatial scale of observations within the video may vary depending on different factors such as underwater visibility (e.g., suspended particulate matter, upwelling events, or light penetration). These factors can influence the effective spatial resolution of observations in a 360° video. To score the video images collected by BRUVS, we used MaxN (the maximum number of individuals of the same species observed together in a single video frame per total minute of footage) [35]. MaxN is one of the most common conservative indexes used for relative abundance in BRUVS analyses because it avoids counting and measuring the same individual more than once [36–39]. We used the tiny planet view (Figure 2c) to score the MaxN values in this study. All individuals were identified to the lowest taxonomic level possible. In addition to species identification and MaxN abundances, the sites were classified based on the predominant habitat and topographic complexity.

Habitat categories were assessed after analysing all videos to compare sites accurately (Table 1). We based our habitat categorisation on the Airoldi and Beck definition [40], where a habitat is identified by the predominant feature contributing to its structure. This structure is derived from biological components, such as vegetation (e.g., macroalgae), or geological features, such as rocky substrates. Considering this, habitat types in our study area encompass sandy bottoms ($n = 59$), rocky reefs ($n = 71$), macroalgae formations ($n = 58$), and mixed habitats (a combination of sandy and rocky reef areas in equal proportion) ($n = 58$) after video analyses classification. To score the topographic complexity, we used the terrain ruggedness index (TRI) since it objectively measures habitat heterogeneity [41]. TRI values were extracted from digital elevation models in Quantum Gis (QGIS) [42] using the algorithm available within the open-source GDAL (Geospatial Data Abstraction Library) tools [43]. This topographic heterogeneity is considered an essential factor in species distribution since when heterogeneity increases, the number of microhabitats and the ability to shelter species also increase [44]. Low TRI values represent relatively flat and homogeneous areas, such as sand and mud, in contrast to high values, which represent rugged and more heterogeneous areas (Table 1).

**Table 1.** Biophysical variables classified in video analysis, including their overall definitions.

| Variable | Estimation Method | Levels and Definition |
|---|---|---|
| Habitat | Categorised during video analysis and based on the underlying habitat structure in the BRUVS field of view. | Rock: sea bottom with bedrock as the underlying substrate. Mix: areas where rocky reefs are intermingled with sand beds in equal proportions. Macroalgae formations: dominance of algae (primarily brown algae). Sand Bed: presence of a sandy substrate as the dominant underlying surface. |
| Topographic complexity (Terrain Ruggedness Index—TRI) | Calculated in software QGIS using GDAL tools. | Continuous; low values represent flat areas; high values represent rugged ones. |

### 2.4. Data Analyses

To ensure that the role of habitat and complexity in the distribution of demersal and benthopelagic species is properly understood, we have excluded species that display predominant pelagic behaviour from our analyses. This measure was taken to avoid any bias in the results. The data matrix was subjected to a square root transformation, and multivariate analysis techniques were applied to determine if and how the demersal and benthopelagic assemblages differ in various habitat types defined by distinct bottom compositions and heterogeneity. Community analyses were performed using a non-parametric permutational multivariate analysis of variance (PERMANOVA) using PRIMER 6.0 software [45]. The test design was based on a two-way model design: habitat (fixed, four levels: Sand vs. Rock vs. Macroalgae formations vs. Mix) and topographic complexity (fixed, continuous). Afterwards, we performed the PERMDISP analysis to assess whether the observed differences could be due to variations in multivariate dispersion related to the centroids' location. A pair-wise test was performed to compare differences between and within groups for pairs of factor levels when appropriate. To investigate patterns in ecological assemblages and community composition, a multivariate analysis technique known as canonical analysis of principal coordinates (CAP) was applied. CAP was used as an exploratory tool to visualise relationships, trends, and dissimilarities among different habitat types. Species richness, abundance, Simpson diversity index, and equitability index were analysed using the vegan package [46] in R software version 4.2.2. To assess variations in these indexes and assemblage characteristics, we employed generalised linear models (GLMs), and models were structured following a two-way design, similar to PERMANOVA, with two factors: habitat (fixed, four levels: Sand vs. Rock vs. Macroalgae formations vs. Mix) and topographic complexity (fixed, continuous). A similarity percentage analysis (SIMPER) was applied to

identify the percentage contribution of each taxon to the Bray–Curtis dissimilarity between habitats and topographic complexity, with a 90% accumulative contribution cut-off point. This analysis allows the emphasis on the species responsible for the significant differences observed in the community.

## 3. Results

### 3.1. Taxonomic Diversity in Coastal Habitats

A total of 11,070 min ($\approx$184 h) of videos were analysed, and 1495 individuals corresponding to 31 species were identified (Table A1, Appendix A). The fauna documented belonged to three phyla (Chordata, Arthropoda, and Mollusca) and 19 families. Overall, the Chordata organisms were the most abundant (95.6% of total abundance) and had the highest number of species identified between habitats (83.8% of total number of species). Of these, the four most abundant species based on the total MaxN values (TMaxN) and mean abundance (mean MaxN $\pm$ SD) were *Trisopterus luscus* with a total of 431 individuals ($2.21 \pm 5.55$), *Diplodus vulgaris* with 310 individuals ($1.58 \pm 5.23$), *Ctenolabrus rupestris* with 150 individuals ($0.76 \pm 1.10$), and *Dicentrarchus labrax* with 94 individuals ($0.48 \pm 1.87$).

The species richness and abundance varied between families in the different habitats. Rock and Mix habitats stood out as the ones that support the highest family diversity, encompassing 13 families each from the three different phyla. A comprehensive assembly of 559 individuals (TMaxN) from 23 species was observed within the Rock habitat, followed by Mix with a TMaxN of 507 individuals identified, representing 22 species from 13 families (across the three phyla). Additionally, we observed TMaxN values of 315 and 114 in Macroalgae and Sand habitats, respectively. Macroalgae formations comprised 19 species from 11 families (Chordata and Mollusca). In comparison, the sand habitat comprised 12 species from 9 families and three identified phyla (see Table 2 for visualisation).

**Table 2.** Family distribution between habitats. Numbers indicate the total species richness within each family.

| Phylum | Family | Habitat | | | |
|---|---|---|---|---|---|
| | | Sand | Macroalgae | Mix | Rock |
| Chordata | Labridae | 4 | 5 | 5 | 5 |
| Chordata | Sparidae | 1 | 3 | 3 | 4 |
| Chordata | Blenniidae | 1 | 2 | 2 | 2 |
| Chordata | Congridae | 1 | 1 | 1 | 1 |
| Chordata | Gadidae | 1 | 1 | 1 | 1 |
| Chordata | Moronidae | 1 | 1 | 1 | 1 |
| Mollusca | Octopodidae | 1 | 1 | 1 | 1 |
| Chordata | Mugilidae | - | 2 | 2 | 2 |
| Chordata | Serranidae | - | 1 | 1 | 1 |
| Arthropoda | Portunidae | 1 | - | 2 | - |
| Chordata | Ammodytidae | - | - | 1 | 2 |
| Chordata | Rajidae | - | 1 | - | - |
| Mollusca | Sepiidae | - | 1 | - | - |
| Arthropoda | Nephropidae | - | - | - | 1 |
| Chordata | Balistidae | - | - | 1 | - |
| Chordata | Physidae | - | - | - | 1 |
| Chordata | Scophthalmidae | - | - | - | 1 |
| Chordata | Triglidae | 1 | - | - | - |
| Chordata | Mullidae | - | - | 1 | - |
| | Species Total | 12 | 19 | 22 | 23 |

### 3.2. Demersal and Benthopelagic Assemblages Composition

Our PERMANOVA analysis revealed significant differences in the community structures of demersal and benthopelagic assemblages across the different habitats, underscored by the statistically significant distinctions associated with habitat type ($p$ = 0.0001, Table 3). Notably, pair-wise comparisons indicated significant differences in the assemblage composition in Macroalgae vs. Sand, Macroalgae vs. Rock, Sand vs. Mix, and between Sand and Rock (Table 4).

**Table 3.** Permutational multivariate analysis of variance (PERMANOVA) of demersal and benthopelagic assemblages. Complexity values are related to the terrain ruggedness index—TRI. Bold letters represent significant differences (* represents the interaction between factors).

| Source | df | SS | MS | Pseudo-F | P (perm) | Unique Perms |
|---|---|---|---|---|---|---|
| Habitat | 3 | 37,232 | 12,411 | 4.3557 | **0.0001** | 9936 |
| Complexity | 66 | $1.8855 \times 10^5$ | 2856.8 | 1.0026 | 0.4828 | 9703 |
| Habitat*Complexity | 50 | $1.5309 \times 10^5$ | 3061.8 | 1.0746 | 0.2527 | 9761 |
| Residual | 75 | $2.137 \times 10^5$ | 2849.3 | | | |
| Total | 194 | $6.1278 \times 10^5$ | | | | |
| PERMDISP (Habitat) | | F: 13.517 | | | P (perm): 0.0001 | |
| PERMDISP (Complexity) | | F: 8.1968 | | | P (perm): 0.0001 | |

**Table 4.** Results of pair-wise tests for demersal and benthopelagic assemblages between habitats. Bold letters represent significant differences.

| | Groups | t | P (Perm) | Unique Perms |
|---|---|---|---|---|
| | Macroalgae, Sand | 1.9386 | **0.0024** | 9942 |
| | Macroalgae, Mix | 1.3927 | 0.0818 | 9937 |
| Habitat | Macroalgae, Rock | 2.2907 | **0.0001** | 9937 |
| | Sand, Mix | 1.8025 | **0.0081** | 9929 |
| | Sand, Rock | 2.839 | **0.0001** | 9934 |
| | Mix, Rock | 0.74279 | 0.7512 | 9950 |

PERMDISP analysis indicates that the dispersion of samples was significant between habitats ($p$ = 0.0001). Similar outcomes occur with topographic complexity, with the PERMDISP results indicating a significant dispersion of replicates among complexity values ($p$ = 0.0001). In other words, the assemblages are different not only in terms of composition but also in terms of how those species are distributed within the habitats. These results show that differences in community composition between habitats may be due to differences in assemblage structure and the variability of those assemblages within the habitats (dispersion). The canonical analysis of principal coordinates (CAP) further substantiates our statistical findings, visually showing the grouping and overlap of assemblages across habitats (Figure 3). According to CAP, habitat arrangement in the multivariate space revealed a separation along the CAP1 axis; most of the consolidated seabed sites are concentrated to the left of the CAP1 axis, and the unconsolidated seabed sites are to the right. The discernible overlap among Rock and Mix habitats indicates a degree of ecological similarity in these sites. In contrast, the separation of most Macroalgae formations and Sand sites suggests a distinctive ecological niche or set of conditions that shape their community structure.

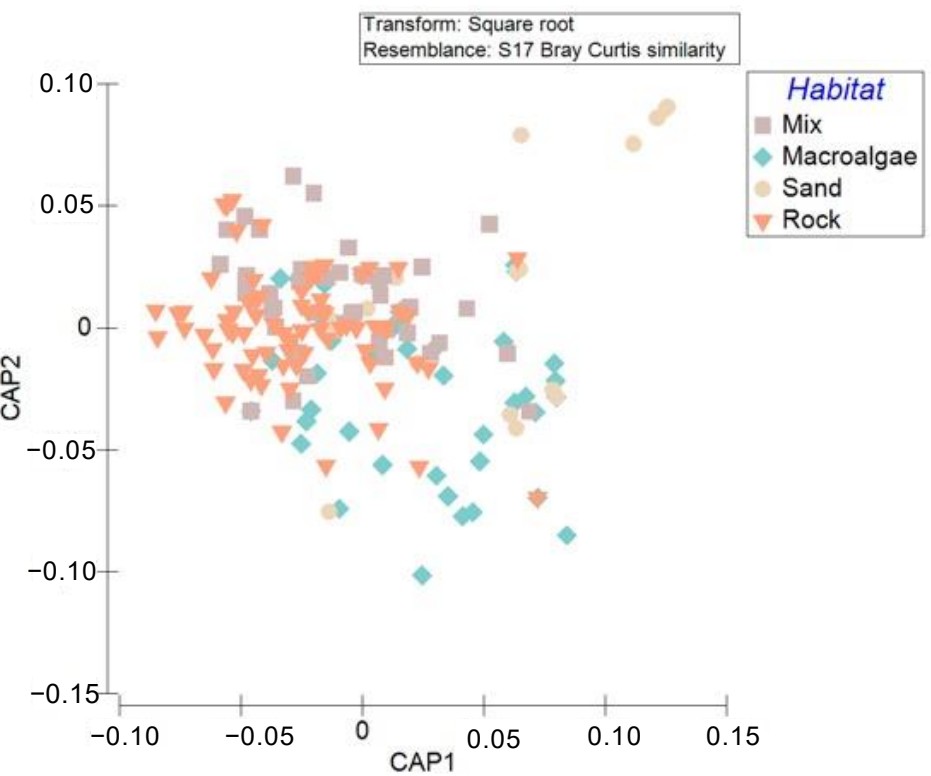

**Figure 3.** Canonical analysis of principal coordinates (CAP) of demersal and benthopelagic assemblages among sites according to habitat type.

### 3.3. Demersal and Benthopelagic Diversity Patterns

GLM analyses were applied to further dissect the influence of habitat and complexity on demersal and benthopelagic species and reveal significant differences between habitats and complexity values (Table A2, Appendix A). Overall, the GLM results, supplemented by boxplot visualisations (Figure 4), revealed nuanced biodiversity patterns across the different habitats. The results indicate a significant effect of habitat type on species richness, with Rock showcasing a significant positive effect on species richness ($Pr(>|z|) = 0.039$) and Sand showing a significant negative impact ($Pr(>|z|) = 0.034$). Additionally, some of the TRI values within the range of the first and second quartiles showcase statistically significant results. In relative abundance, significant positive differences were also revealed in Rock habitats ($Pr(>|z|) = 0.003$). Tukey tests also showed significant differences between Rock vs. Macroalgae formations ($Pr(>|z|) = 0.017$), Sand vs. Mix ($Pr(>|z|) = 0.043$), and Sand vs. Rock ($Pr(>|z|) = 0.002$). TRI displayed the same pattern as species richness, where some levels of the first and second quartiles of the terrain ruggedness index showcase statistically significant results. In the Simpson diversity index, Sand displayed negative statistical differences ($Pr(>|z|) = 0.001$), and Rock, on the other hand, displayed positive influence and statistical differences ($Pr(>|z|) = 0.005$). Between groups, statistical differences were found between Mix vs. Sand ($Pr(>|z|) < 0.009$) and Sand vs. Rock ($Pr(>|z|) < 0.001$). TRI presents a similar repetition of the previous analyses, but in the Simpson diversity, the highest TRI value also showcased significant differences. Finally, GLMs on the evenness component of the Simpson diversity index did not reveal significant differences in evenness across the various habitat types ($Pr(>|z|) > 0.05$), and in TRI, only two values between the first and second quartiles, present significant differences (see Table A2, for completed GLM results). These results suggest that while the overall diversity in the Simpson index may vary across habitats, the evenness component of this diversity—how evenly individual species are represented within each habitat—does not significantly differ among the four habitats, presenting only differences in some TRI values (heterogeneity) (Figure 4). These results show that, while there may be minor variations in the evenness of species distri-

butions within each habitat, these differences are insufficient to be considered statistically significant. In ecological terms, this suggests that the habitats, although differing in species richness and abundance, maintain a balance in terms of how individual species contribute to the community without any single species dominating excessively over others.

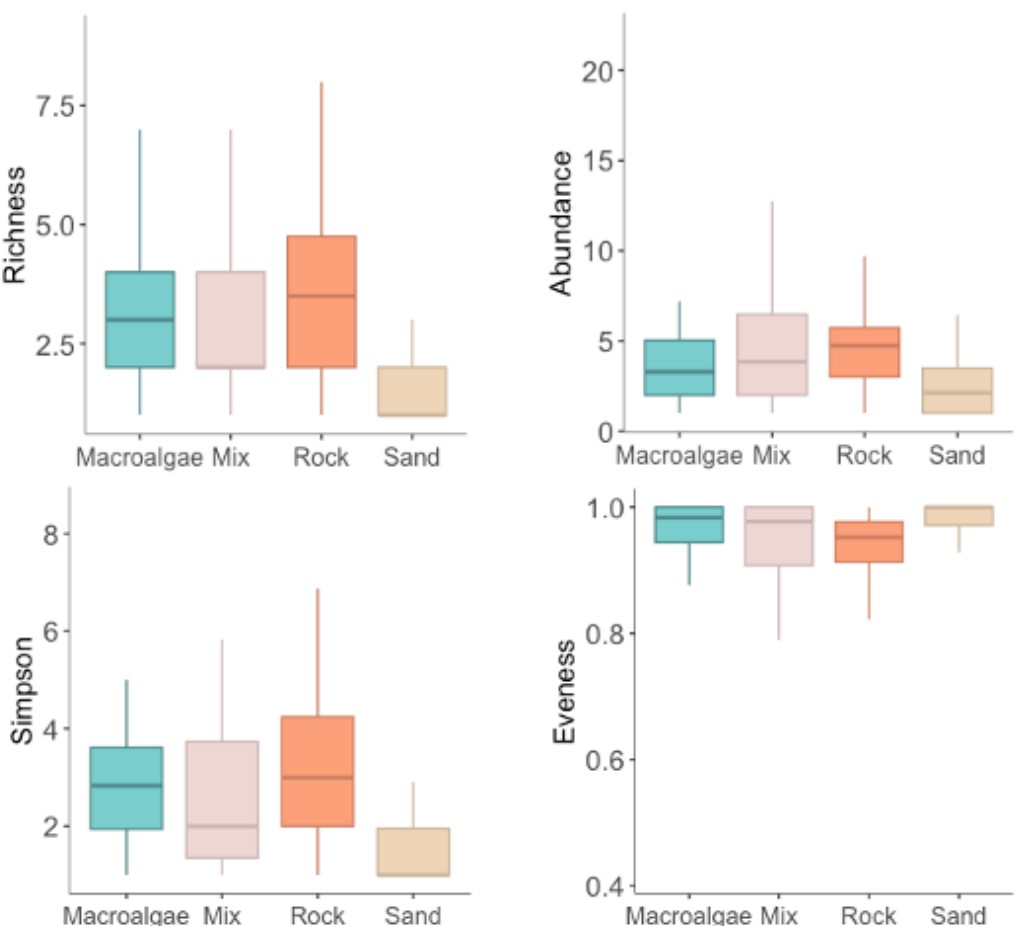

**Figure 4.** Mean species richness, relative abundance of individuals (MaxN), Simpson diversity index and equitability index of demersal, and benthopelagic assemblages in the different habitats.

SIMPER results showed that the dissimilarity in demersal and benthopelagic assemblages between Macroalgae formations and Sand averaged 86%. *Carcinus maenas* contributed with 15.07%, *Trisopterus luscus* with 14.92%, *Diplodus vulgaris* with 14.76%, and *Dicentrarchus labrax* with 12.30% to this dissimilarity. Macroalgae formations and Rock also presented higher levels of dissimilarity (77.09%), and within this dissimilarity, *Trisopterus luscus* played a prominent role with a contribution of 17.24%, *Diplodus vulgaris* followed by 14.03%, *Ctenolabrus rupestris* with 12.97%, *Dicentrarchus labrax* with 8.61%, and *Labrus bergylta* with 8.58%. Between Sand and Mix, the dissimilarity was 86.60%; *Trisopterus luscus* (21.98%), *Carcinus maenas* (14.20%), *Diplodus vulgaris* (12.78%), and *Ctenolabrus rupestris* (8.99%) were responsible for 57.95% of it. Finally, the dissimilarity between Sand and Rock in the demersal and benthopelagic assemblages reached 85.018%. Significant contributors to this dissimilarity were *Trisopterus luscus* (18.66%), *Ctenolabrus rupestris* (12.97%), *Diplodus vulgaris* (12.88%), and *Carcinus maenas* (12.77%) (Table 5).

**Table 5.** SIMPER analysis results for the top contributors of the dissimilarity of demersal and benthopelagic assemblages between habitats as determined by PERMANOVA based on the four root MaxN abundances data.

| Species | Average Abundance | | Av.Diss | Diss/SD | Contrib% | Cum% |
|---|---|---|---|---|---|---|
| | **Macroalgae** | **Sand** | | | | |
| *Carcinus maenas* | 0.00 | 0.74 | 12.96 | 0.61 | 15.07 | 15.07 |
| *Trisopterus luscus* | 0.60 | 0.46 | 12.83 | 0.90 | 14.92 | 29.99 |
| *Diplodus vulgaris* | 0.85 | 0.41 | 12.69 | 1.00 | 14.76 | 44.75 |
| *Dicentrarchus labrax* | 0.53 | 0.31 | 10.58 | 0.80 | 12.30 | 57.05 |
| *Symphodus melops* | 0.25 | 0.04 | 5.61 | 0.51 | 6.52 | 63.57 |
| *Ctenolabrus rupestris* | 0.27 | 0.13 | 5.18 | 0.57 | 6.03 | 69.60 |
| *Symphodus bailloni* | 0.27 | 0.06 | 4.93 | 0.51 | 5.73 | 75.33 |
| *Labrus bergylta* | 0.23 | 0.13 | 4.76 | 0.55 | 5.53 | 80.86 |
| *Parablennius gattorugine* | 0.16 | 0.04 | 2.82 | 0.44 | 3.27 | 84.13 |
| *Chelon auratus* | 0.13 | 0.00 | 2.16 | 0.29 | 2.51 | 86.64 |
| *Chelidonichthys lucerna* | 0.00 | 0.08 | 1.84 | 0.27 | 2.14 | 88.78 |
| *Conger conger* | 0.05 | 0.04 | 1.61 | 0.26 | 1.88 | 90.66 |
| | **Macroalgae** | **Rock** | | | | |
| *Trisopterus luscus* | 0.60 | 1.16 | 13.29 | 1.11 | 17.24 | 17.24 |
| *Diplodus vulgaris* | 0.85 | 0.73 | 10.81 | 1.03 | 14.03 | 31.27 |
| *Ctenolabrus rupestris* | 0.27 | 0.81 | 10.00 | 1.03 | 12.97 | 44.24 |
| *Dicentrarchus labrax* | 0.53 | 0.13 | 6.64 | 0.76 | 8.61 | 52.85 |
| *Labrus bergylta* | 0.23 | 0.40 | 6.08 | 0.73 | 7.89 | 60.74 |
| *Symphodus bailloni* | 0.27 | 0.31 | 5.34 | 0.70 | 6.93 | 67.67 |
| *Serranus cabrilla* | 0.05 | 0.39 | 4.83 | 0.73 | 6.27 | 73.93 |
| *Parablennius gattorugine* | 0.16 | 0.27 | 4.33 | 0.63 | 5.61 | 79.54 |
| *Symphodus melops* | 0.25 | 0.06 | 4.23 | 0.52 | 5.49 | 85.04 |
| *Chelon auratus* | 0.13 | 0.04 | 1.97 | 0.33 | 2.55 | 87.59 |
| *Conger conger* | 0.05 | 0.08 | 1.55 | 0.32 | 2.01 | 89.60 |
| *Coris julis* | 0.05 | 0.07 | 1.27 | 0.31 | 1.65 | 91.25 |
| | **Sand** | **Mix** | | | | |
| *Trisopterus luscus* | 0.46 | 1.36 | 19.04 | 1.02 | 21.98 | 21.98 |
| *Carcinus maenas* | 0.74 | 0.02 | 12.29 | 0.60 | 14.20 | 36.18 |
| *Diplodus vulgaris* | 0.41 | 0.68 | 11.07 | 0.85 | 12.78 | 48.96 |
| *Ctenolabrus rupestris* | 0.13 | 0.57 | 7.79 | 0.79 | 8.99 | 57.95 |
| *Dicentrarchus labrax* | 0.31 | 0.22 | 7.39 | 0.56 | 8.53 | 66.49 |
| *Labrus bergylta* | 0.13 | 0.32 | 6.16 | 0.58 | 7.11 | 73.60 |
| *Serranus cabrilla* | 0.00 | 0.29 | 4.05 | 0.53 | 4.68 | 78.27 |
| *Symphodus bailloni* | 0.06 | 0.18 | 3.40 | 0.44 | 3.93 | 82.20 |
| *Coris julis* | 0.00 | 0.22 | 3.12 | 0.41 | 3.61 | 85.81 |
| *Chelidonichthys lucerna* | 0.08 | 0.00 | 1.73 | 0.26 | 1.99 | 87.80 |
| *Parablennius gattorugine* | 0.04 | 0.10 | 1.55 | 0.33 | 1.79 | 89.59 |
| *Mullus surmuletus* | 0.00 | 0.14 | 1.52 | 0.31 | 1.76 | 91.35 |
| | **Sand** | **Rock** | | | | |
| *Trisopterus luscus* | 0.46 | 1.16 | 15.89 | 1.18 | 18.66 | 18.66 |
| *Ctenolabrus rupestris* | 0.13 | 0.81 | 11.05 | 1.03 | 12.97 | 31.63 |
| *Diplodus vulgaris* | 0.41 | 0.73 | 10.97 | 0.90 | 12.88 | 44.51 |
| *Carcinus maenas* | 0.74 | 0.00 | 10.88 | 0.60 | 12.77 | 57.28 |
| *Labrus bergylta* | 0.13 | 0.40 | 6.59 | 0.71 | 7.73 | 65.01 |
| *Dicentrarchus labrax* | 0.31 | 0.13 | 5.83 | 0.56 | 6.84 | 71.85 |
| *Serranus cabrilla* | 0.00 | 0.39 | 5.33 | 0.72 | 6.26 | 78.11 |
| *Symphodus bailloni* | 0.06 | 0.31 | 4.36 | 0.61 | 5.12 | 83.23 |
| *Parablennius gattorugine* | 0.04 | 0.27 | 4.00 | 0.55 | 4.69 | 87.92 |
| *Conger conger* | 0.04 | 0.08 | 1.50 | 0.34 | 1.77 | 89.69 |
| *Chelidonichthys lucerna* | 0.08 | 0.00 | 1.49 | 0.27 | 1.75 | 91.44 |

## 4. Discussion

The mosaic seabed/habitat and biodiversity patterns unveiled by our study reflect a dynamic interplay between habitat complexity and the distribution of demersal and benthopelagic species in the coastal area of Viana do Castelo—NW Iberian coast. Our findings reveal substantial differences in the demersal and benthopelagic assemblages, species richness, MaxN abundances, and diversity indices among the habitat's complexity. The results of this study set the baseline data and highlight the importance of coastal heterogeneous mosaic seascapes in supporting the diversity of demersal and benthopelagic species as well as the importance of rocky seabeds in supporting the capacity of these assemblages.

### 4.1. Diversity of the Demersal and Benthopelagic Species in Coastal Habitats

One of the most marked findings in our study is the different patterns in species richness and abundance noted across the habitats presented in the study area. The Rock habitat (the most prevalent habitat on our mosaic seabed), displayed the highest diversity, with 23 species identified from 13 families and the highest mean MaxN abundance (4.70 ± 2.24 organisms). Mix habitats presented similar patterns with 22 species from 13 families and a mean MaxN abundance of (4.58 ± 3.44). Macroalgae formations comprised 19 species from 11 families and lower mean MaxN abundances (3.86 ± 3.36) than rocky environments. These numbers suggest that the heterogeneity and the presence of hard substrate contribute to creating habitats with different niches supporting biodiversity, as reported by Flávio et al. [47]. With their simpler structure, sand habitats presented the lowest species richness (12 species belonging to 9 families) and the lowest mean MaxN abundance (2.52 ± 1.58). Our results emphasise the crucial role of habitat structural complexity in supporting marine life. Structured environments with hard substrates, like Rock and Mix beds, provide essential resources and refuges for a variety of species due to their structure, as seen in previous studies [48–50].

### 4.2. Influence of Habitat Complexity on Biodiversity

The combined statistical analyses performed in this study provide detailed insights into biodiversity composition within and between various habitats. These findings offer a crucial understanding of habitat conservation and inform future effective biodiversity management strategies. The PERMANOVA results (Table 2) indicated significant differences in community composition that are influenced by habitat type ($p = 0.0001$) but not by terrain ruggedness index (TRI) that composed the topographic complexity in the area ($p > 0.05$). For instance, the significative differences found in the pair-wise results (Table 3) between Macroalgae formations vs. Sand, Macroalgae formations vs. Rock, Sand vs. Mix, and Rock vs. Sand could be the reflection of distinct physical structures (e.g., the presence of hard subtract and heterogeneity) and biological interactions within these habitats. These differences highlight the fundamental ecological principle that biodiversity is often a function of a habitat's physical structure and complexity. The PERMDISP analyses added a layer of nuance to our understanding by revealing variability in these communities within habitats. The significant differences in dispersion patterns (Table 2) suggest that habitat structure, intra-spatial arrangement, and interactions influence the communities within habitats. This variability could be related to specific habitat characteristics, different spatial arrangements that provide diverse ecological niches, or differential responses to biotic factors such as predation or competition [51]. Various microhabitats can influence the dispersion of the assemblages' intra-habitats even within a broadly defined habitat. For example, one of the rock habitat sites may present crevices of different sizes, boulders, or vegetation cover, providing different conditions and resources (e.g., various shelters and more food resources), than a more naked and compact rock. The variation in microhabitats can lead to a more biodiverse assemblage at different sites within the exact broad habitat characterisation. Different studies have already demonstrated that species distribution is not random within areas that are often perceived as homogeneous habitats. Instead, it closely aligns with specific habitat patterns, which can be found at micro-scales [49,52,53].

For instance, habitats with diverse intra-spatial arrangements may reduce competition by offering different competitive refuges or a broader range of resources [54]. Additionally, they can reduce predation risks by providing numerous shelters for prey and reducing the frequency of predator–prey encounters [55]. Also, structurally complex habitats have been shown to lessen predation across various areasand are more utilised by prey species when predators are present [56–59]. It is also crucial to consider the existence of dynamic ecological processes and the fluctuation of species between habitats. Sand areas closer to rocks can present different assemblages compared to those inside long sand bank platforms. Considering that surveys inside the same habitats were conducted on different days, we cannot discard the variations in environmental factors such as light, current, or sedimentation, which can also affect species assemblages. The observed overlaps in the CAP analysis (Figure 3) among habitats with more hard subtract structures, such as rocky and mixed substrates, suggest a degree of ecological similarity, shared species assemblage, or equivalency in habitat provision. Contrariwise, the distinct separation of sand habitats indicates a unique assemblage, potentially driven by the homogeneity and softness of the substrate, supporting fewer microhabitats. GLM results further dissected these relationships by quantifying and demonstrating significant differences in the habitats on species richness, abundance, and Simpson diversity, indicating that habitat and some medium levels of heterogeneity (TRI) are determinants in demersal and benthopelagic species distribution (see Figure 4 and Table A2).

Interestingly, the lack of significant differences in species evenness across habitats (Figure 4) suggests a balance in species representation in each habitat, indicative of effective dispersal mechanisms or life-history strategies that allow species to exploit habitats equitably. This evenness is crucial for the stability of marine ecosystems, as it prevents dominance by a particular species and maintains a balance in the community structure. Together, these analyses provide a multifaceted understanding of the ecological patterns across habitats. Incorporating the findings from the SIMPER analysis allows us to pinpoint the species that are most influential in driving the dissimilarity between habitats. The observed dissimilarity in assemblages and the notable contributions of certain species to this dissimilarity highlight the role of these species as potential indicators of habitat quality and assemblage health. They may function as 'keystone species', shaping habitat communities and influencing the presence and abundance of other species through their biological activities.

Species such as pouting *Trisopterus luscus*, common two-banded seabream *Diplodus vulgaris*, and goldsinny *Ctenolabrus rupestris* were key contributors to the observed dissimilarities between habitats, indicative of their specific habitat preferences and ecological roles. When analysing the average abundance of these species between habitats, it is evident that they are more abundant in hard substrate habitats. These results are in line with other studies that have demonstrated the positive association of these species, like the goldsinny *Ctenolabrus rupestris*, with higher structured areas [21,50,60], which have different refuges [21,61]. Also, their attraction to hard-bottom substrate in shallow waters is well documented [50,62,63]. Several reasons can explain the attraction to these complex structured rocky areas, including higher food availability and the existence of more secure shelters that allow them to avoid predation and find refuge, preferentially crevices on rock faces, or between boulders, with two or more entrances [21,61]. Also, since goldsinny is considered a relatively weak swimmer, individuals remain close to protective rocky structures, exhibiting greater site fidelity [60]. This preference is likely driven by the increased predation risk of crossing large sandy areas [64–66]. The pout, *Trisopterus luscus*, also a key benthopelagic species in cold waters [67], often uses coastal zones as nurseries [68–70]. Our findings align with previous research showing higher population densities in areas with hard substrates [71]. This understanding is vital, considering the species' significance for artisanal fishing across Europe, including our study area [72]. Such insights are critical for the sustainable management and habitat conservation of this species. Furthermore, considering the broader ecological context, most identified species occupy an intermediate

position in the food web, crucial in maintaining ecological balance. This food web position makes them vulnerable to change and severely impacts ecological processes, as Olsen et al. [73] highlighted, emphasising the importance of our study's findings for a broader environmental perspective.

On the other hand, focusing, for example, on the average abundance of green crab *Carcinus maenas* (Table 4), our findings indicate a higher association of this species with sandy habitats (Table 5). Similarly, tub gurnard *Chelidonichthys lucerna* also demonstrates a preference for sandy environments. These results suggest that *Carcinus maenas* and tub gurnard *Chelidonichthys lucerna* exhibit similar habitat preferences, highlighting the importance of sandy habitats for these species in the study area. The seabass *Dicentrarchus labrax* also appears related to sand environments, appearing in shallow waters, usually with fine sandy or muddy bottoms [74]. Different results regarding the green crab *Carcinus maenas* were found by Moksnes [75], who concluded that different life stages of crabs were significantly less abundant in open sandy areas compared to regions with more complex habitats. However, this study also stated that habitat-specific predation may have affected these results, which can also explain our results. A deeper understanding of late young species stages adapted to their habitat choice is also crucial to comprehend the species dynamics. For example, in initial life stages, these species may prefer more structured habitats to avoid predation. This knowledge could lead to enhanced management of the harvested populations and of the crucial nursery habitats they depend on in the study area.

*4.3. Conservation Implications and Future Research Directions*

The findings of this study align with and extend the body of work that emphasises the significance of habitat mosaics in coastal environments [15,17,76]. Additionally, they are consistent with other studies that have reported a similar trend of enhanced species richness and abundance in structured habitats across different climatic regions [77–79]. Moreover, our results emphasise the crucial role of rocky substrates' presence within these coastal habitat mosaics. A variety of species are supported by structurally complex habitats formed by rocky reefs, which are also the main providers of physical three-dimensional features in coastal areas in temperate regions [18,78,80].

These insights provide valuable information that can inform effective marine conservation and management strategies, particularly in alignment with the objectives (e.g., articles 8.1a, 9, 10, and 11) of the European Marine Strategy Framework Directive (MSFD; 2008/56/EC), which acknowledges the protection and conservation of European Marine waters and emphasises the importance of safeguarding benthic ecosystems [81].

The site fidelity of some species with economic value should also be considered in coastal fishery management. Species with strong tendencies to remain in specific locations, combined with their limited movement range, make demersal and benthopelagic populations particularly susceptible to the negative impact of local fishing efforts [82].

Although our study is a valuable source of insights, certain limitations must be acknowledged. The interpretation of topographic indices such as the terrain ruggedness index (TRI) must be cautiously considered since it depends on the accuracy and resolution of digital elevation models. Also, the study's seasonal scope might overlook temporal variations in species assemblages. Since this is the first study to comprehensively examine the influence of habitat and topographic complexity on species distribution in this region, future studies should expand upon our temporal range even during challenging survey conditions and in developed digital elevation models with higher accuracy and resolution. With more detailed digital elevation models, higher precision can be achieved in accessing the influence of habitat and complexity on species, since, on smaller scales, heterogeneity in sediment was determined as a key factor for species diversity [83].

Exploring habitats' spatial arrangement and interconnectivity will deepen our knowledge of their influence on species assemblages. Assessing the individual and combined impacts of different habitats on ecosystem functionality is vital for planning effective conservation strategies, especially given the growing pressures on coastal ecosystems. One

fundamental step for the future is to associate ecological studies with geological analyses. Understanding which types of rock (e.g., granite or shale) compose the seabed mosaic and how these influence the communities is necessary to correctly manage coastal areas. Compact rocks like granite and quartzite may have a lower availability of shelters than shales, which are more prone to forming spaces that can easily be used as hiding places. This information would enhance the baseline data we have provided through this study, offering further insights into the marine biodiversity of the NW Iberian coast.

However, these limitations did not bias this study's interpretation of how habitats and heterogeneity affect the demersal and benthopelagic distributions. Overall, we can emphasise the complex interplay between habitat heterogeneity and the ecological niches they provide. Hard-bottom habitats, such as mixed and rocky ones, support greater species diversity and exhibit higher abundance levels. The uniformity and lower structural complexity of sand habitats may limit their capacity to support a similar range of species in terms of richness and abundance. Each region's unique interplay between habitats and the structures they provide influences the presence and abundance of specific species. Understanding the existing habitat mosaics and identifying predominant, economically significant species in these regions is crucial. These data are fundamental to comprehending these species' status, habitat interactions, and necessary conservation measures. From a conservation perspective, the findings of this study highlight the need for more regional studies and habitat-specific management strategies. Protecting structurally complex hard-bottom habitats is vital to maintaining high levels of marine biodiversity, particularly in areas where the complexity and connectivity of different habitat types converge. Identifying and preserving these critical regions are necessary to safeguard marine biodiversity and ensure the sustainability of ecological processes. Furthermore, recognising critical species contributing to community differences provides a targeted approach for monitoring and managing coastal marine biodiversity locally, as changes in the abundance or distribution of these species could indicate shifts in habitat quality or ecosystem health.

## 5. Conclusions

This research contributes to the growing evidence that habitat heterogeneity is a crucial driver of biodiversity in coastal ecosystems. This is the first study examining the influence of habitat and complexity on the demersal and benthopelagic species identified along Viana do Castelo on the NW Iberian coast. By providing basic knowledge about this influence, this study can be used as baseline data about demersal and benthopelagic assemblages on the Iberian coast. The structural features offered within and between habitats play a crucial role in shaping these patterned assemblages, with implications for biodiversity conservation, habitat management, and the resilience response of marine ecosystems to environmental changes. By offering a clearer understanding of how different habitat structures support marine life, our findings provide valuable information that can inform sustainable management and conservation strategies, ensuring the long-term health and productivity of these vital ecological systems.

**Author Contributions:** Conceptualisation, M.A.G., J.S.T. and P.T.G.; methodology, M.A.G., J.S.T., C.M.A., F.F. and P.T.G.; formal analysis, M.A.G., J.S.T. and P.T.G.; investigation, M.A.G., J.S.T. and P.T.G.; resources, M.A.G., J.S.T. and P.T.G.; data curation, M.A.G.; writing—original draft preparation, M.A.G.; writing—review and editing, M.A.G., J.S.T., C.M.A., F.F. and P.T.G.; visualisation, M.A.G.; supervision, J.S.T. and P.T.G.; project administration, J.S.T. and P.T.G.; funding acquisition, M.A.G., J.S.T. and P.T.G. All authors have read and agreed to the published version of the manuscript.

**Funding:** This study was supported by the "Contrato-Programa" UIDB/04050/2020 funded by national funds through the FCT I.P. and through LA/P/0069/2020 granted to the Associate Laboratory ARNET. Financial support granted by the FCT to MAG (PD/BD/143088/2018 and COVID/BD/153031/2022) and C.M.A (PD/BD/150365/2019) is also acknowledged.

**Institutional Review Board Statement:** Not applicable.

**Informed Consent Statement:** Not applicable.

**Data Availability Statement:** The data presented in this study are available upon request from the corresponding author. The data are not publicly available because this data set may be included as part of other ongoing studies.

**Conflicts of Interest:** The authors declare no conflict of interest. The funders had no role in the design of the study; in the collection, analyses, or interpretation of data; in the writing of the manuscript; or in the decision to publish the results.

## Appendix A

**Table A1.** List of demersal and benthopelagic species identified in the habitat mosaic of Viana do Castelo, NW Iberian coast (in descending order from most abundant to least abundant).

| Phylum | Family | Specie |
|--------|--------|--------|
| Chordata | Gadidae | *Trisopterus luscus* |
| Chordata | Sparidae | *Diplodus vulgaris* |
| Chordata | Labridae | *Ctenolabrus rupestris* |
| Chordata | Moronidae | *Dicentrarchus labrax* |
| Chordata | Ammodytidae | *Ammodytes tobianus* |
| Chordata | Labridae | *Labrus bergylta* |
| Arthropoda | Portunidae | *Carcinus maenas* |
| Chordata | Labridae | *Symphodus bailloni* |
| Chordata | Serranidae | *Serranus cabrilla* |
| Chordata | Blenniidae | *Parablennius gattorugine* |
| Chordata | Sparidae | *Diplodus sargus* |
| Chordata | Labridae | *Coris julis* |
| Chordata | Labridae | *Symphodus melops* |
| Chordata | Mugilidae | *Chelon auratus* |
| Chordata | Congridae | *Conger conger* |
| Chordata | Mugilidae | *Chelon labrosus* |
| Chordata | Mullidae | *Mullus surmuletus* |
| Mollusca | Octopodidae | *Octopus vulgaris* |
| Chordata | Sparidae | *Spondyliosoma cantharus* |
| Chordata | Blenniidae | *Parablennius pilicornis* |
| Chordata | Sparidae | *Sarpa salpa* |
| Chordata | Triglidae | *Chelidonichthys lucerna* |
| Chordata | Phycidae | *Phycis phycis* |
| Arthropoda | Portunidae | *Necora puber* |
| Chordata | Ammodytidae | *Hyperoplus lanceolatus* |
| Chordata | Scophthalmidae | *Zeugopterus punctatus* |
| Arthropoda | Nephropidae | *Homarus gammarus* |
| Chordata | Rajidae | *Raja undulata* |
| Mollusca | Sepiidae | *Sepia officinalis* |
| Chordata | Sparidae | *Diplodus cervinus* |
| Chordata | Balistidae | *Balistes capriscus* |

**Table A2.** Fixed effects for the generalised linear model (GLM) for functional species richness, abundance, Simpson diversity, and evenness. Only significant TRI are represented in the table.

|  | Estimate Std. | Error z | Z Value | Pr (>|z|) |
|--|---------------|---------|---------|-----------|
| Species richness |  |  |  |  |
| Intercept | 0.30250 | 0.50691 | 0.597 | 0.5507 |
| Habitat sand | −0.43151 | 0.20374 | −2.118 | **0.0342** |
| Habitat mix | 0.05181 | 0.15057 | 0.344 | 0.7308 |
| Habitat rock | 0.26723 | 0.13000 | 2.056 | **0.0398** |
| TRI 1.12 | 1.14071 | 0.56890 | 2.005 | **0.0449** |
| TRI 2.62 | 1.30535 | 0.56406 | 2.314 | **0.0207** |
| TRI 3.75 | 1.27210 | 0.54090 | 2.352 | **0.0187** |
| TRI 8.125 | 1.50971 | 0.61659 | 2.449 | **0.0143** |

**Table A2.** *Cont.*

|  | Estimate Std. | Error z | Z Value | Pr (>\|z\|) |
|---|---|---|---|---|
| Pair-wise habitat |  |  |  |  |
| sand-algae | −0.43151 | 0.20374 | −2.118 | 0.14277 |
| mix-algae | 0.05181 | 0.15057 | 0.344 | 0.98559 |
| rock-algae | 0.26723 | 0.13000 | 2.056 | 0.16272 |
| mix-sand | 0.48332 | 0.20807 | 2.323 | 0.08950 |
| rock-sand | 0.69874 | 0.20339 | 3.435 | **0.00309** |
| rock-mix | 0.21541 | 0.15087 | 1.428 | 0.47421 |
| Abundance |  |  |  |  |
| Intercept | 0.57574 | 0.37098 | 1.552 | 0.123209 |
| Habitat sand | −0.22525 | 0.17066 | −1.320 | 0.189301 |
| Habitat mix | 0.24731 | 0.14804 | 1.671 | 0.097306 |
| Habitat rock | 0.38228 | 0.13024 | 2.935 | **0.003968** |
| TRI 1.12 | 1.81298 | 0.47500 | 3.817 | **0.000212** |
| TRI 2.62 | 0.96752 | 0.46352 | 2.087 | **0.038887** |
| TRI 3.75 | 1.31991 | 0.43299 | 3.048 | **0.002809** |
| TRI 5.75 | 0.85355 | 0.42902 | 1.990 | **0.048826** |
| TRI 7.625 | 1.29056 | 0.64736 | 1.994 | **0.048375** |
| TRI 8.12 | 1.30965 | 0.64696 | 2.024 | **0.045069** |
| Pair-wise habitat |  |  |  |  |
| sand-algae | −0.2252 | 0.1707 | −1.320 | 0.54661 |
| mix-algae | 0.2473 | 0.1480 | 1.671 | 0.33584 |
| rock-algae | 0.3823 | 0.1302 | 2.935 | **0.01712** |
| mix-sand | 0.4726 | 0.1803 | 2.621 | **0.04310** |
| rock-sand | 0.6075 | 0.1723 | 3.526 | **0.00232** |
| rock-mix | 0.1350 | 0.1475 | 0.915 | 0.79443 |
| Simpson diversity |  |  |  |  |
| Intercept | 0.260279 | 0.269267 | 0.967 | 0.335600 |
| Habitat sand | −0.436706 | 0.129872 | −3.363 | **0.001025** |
| Habitat mix | −0.002609 | 0.117799 | −0.022 | 0.982366 |
| Habitat rock | 0.291935 | 0.103534 | 2.820 | **0.005591** |
| TRI 0.75 | 0.947355 | 0.350629 | 2.702 | **0.007853** |
| TRI 1 | 0.948522 | 0.372574 | 2.546 | **0.012116** |
| TRI 1.12 | 0.921093 | 0.372186 | 2.475 | **0.014670** |
| TRI 1.25 | 0.928816 | 0.372287 | 2.495 | **0.013904** |
| TRI 2 | 0.886297 | 0.309524 | 2.863 | **0.004916** |
| TRI 2.25 | 1.039249 | 0.334634 | 3.106 | **0.002350** |
| TRI 2.37 | 0.727429 | 0.306913 | 2.370 | **0.019311** |
| TRI 2.5 | 1.020710 | 0.372287 | 2.742 | **0.007009** |
| TRI 2.62 | 1.372803 | 0.351153 | 3.909 | **0.000151** |
| TRI 3.125 | 0.798455 | 0.323704 | 2.467 | **0.014994** |
| TRI 3.25 | 1.094760 | 0.375573 | 2.915 | **0.004217** |
| TRI 3.5 | 0.717641 | 0.348111 | 2.062 | **0.041325** |
| TRI 3.75 | 1.208433 | 0.328587 | 3.678 | **0.000348** |
| TRI 4.375 | 0.852410 | 0.375573 | 2.270 | **0.024944** |
| TRI 4.75 | 1.157256 | 0.528916 | 2.188 | **0.030531** |
| TRI 5 | 1.040476 | 0.418589 | 2.486 | **0.014251** |
| TRI 5.75 | 0.981637 | 0.323165 | 3.038 | **0.002904** |
| TRI 7.375 | 1.349159 | 0.526761 | 2.561 | **0.011618** |
| TRI 8.125 | 1.498214 | 0.526976 | 2.843 | **0.005221** |
| TRI 8.625 | 1.375141 | 0.526976 | 2.609 | **0.010174** |
| TRI 19.25 | 1.294577 | 0.532970 | 2.429 | **0.016563** |

**Table A2.** *Cont.*

|  | Estimate Std. | Error z | Z Value | Pr (>\|z\|) |
|---|---|---|---|---|
| **Pair-wise habitat** |  |  |  |  |
| sand-algae | −0.436706 | 0.129872 | −3.363 | 0.00417 |
| mix-algae | −0.002609 | 0.117799 | −0.022 | 1.00000 |
| rock-algae | 0.291935 | 0.103534 | 2.820 | 0.02461 |
| mix-sand | 0.434097 | 0.139524 | 3.111 | **0.00969** |
| rock-sand | 0.728641 | 0.132459 | 5.501 | **<0.001** |
| rock-mix | 0.294544 | 0.118241 | 2.491 | 0.06038 |
| **Evenness** |  |  |  |  |
| Intercept | $-2.715 \times 10^{2}$ | $5.59 \times 10^{-2}$ | −0.486 | 0.628065 |
| Habitat sand | $1.542 \times 10^{-2}$ | $2.69 \times 10^{-2}$ | 0.572 | 0.568514 |
| Habitat mix | $-2.252 \times 10^{-2}$ | $2.44 \times 10^{-2}$ | −0.921 | 0.358976 |
| Habitat rock | $-3.808 \times 10^{-2}$ | $2.15 \times 10^{-2}$ | −1.771 | 0.078954 |
| TRI 1.125 | $-2.295 \times 10^{-1}$ | $7.72 \times 10^{-2}$ | −2.969 | **0.003578** |
| TRI 5.375 | $-2.989 \times 10^{-1}$ | $8.84 \times 10^{-1}$ | −3.378 | **0.000975** |

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
