# Peer review of "Untangling Coastal Diversity: How Habitat Complexity Shapes Demersal and Benthopelagic Assemblages in NW Iberia"

_jmse, doi:10.3390/jmse12040538_

Round 1

Reviewer 1 Report

Comments and Suggestions for Authors

Dear Authors,

The manuscript, focused on the understanding of species-habitat relationships in the coastal waters of Viana do Castelo (Portugal), is well written and informative, featuring appropriate scientific research methods and analyses. The results and conclusions are systematically organized and scientifically supported. The literature review is comprehensive and consistent with the in-text citations in the manuscript.

My only remark is for the section 4.1: Diversity of the demersal and benthopelagic diversity in coastal habitats, please remove the repeated usage of "diversity"; also in this section the abundance of species is indicated on lines 307, 309, and 314 without the inclusion of the unit of measurement.

Reviewer 2 Report

Comments and Suggestions for Authors

jmse-2910558 Gomes et al 2024

A very interesting article and one which does set out some framework for management and conservation.

My main criticism would be that it should be made plain that the comments and evaluations apply to the larger (mobile and predatory/scavenger?) epifauna and consequently many taxa e.g. Mollusca and Polychaeta are under-reported. It might almost be preferable to present it as fish alone.

I should also have expected some comments on scale, both in terms of the heterogeneity of the habitats sampled and consequently the patterns of diversity. See also their own comments lines 436-9 which would be better brought up into the Intro and addressed also in the Methodology and data analysis.

Some smaller points are detailed below. 

Line

Comment

136

Can you give some idea of the area of a frame?

142-52

See comment above on scale. E.g. how large does a habitat have to be for designation as mix?

365-9

An alternative explanation would be simply that the same balance of species are attracted, possibly from neighbouring habitats, by the bait. If you enter distance to nearest other habitat into the analysis, does your contention of balance still hold? See also comments above on scale.

403

Some caution may be needed here, both in terms of the visibility of C. maenas (which would be easier on sediments than under an algal canopy) and their size (smaller ones may prefer cover). In my diving experience I have found many more C. maenas in rocks/algae than on open sands or muds.

Overall, perhaps they should be a bit more cautious about some of the more sweeping claims and conclusions, but I feel that the comments above might quickly be addressed and the ms needs no more than minor changes.
